# Measuring cancer driving force of chromosomal aberrations through multi-layer Boolean implication networks

Ilaria Cosentini[1], Daniele Filippo Condorelli[2], Giorgio Locicero[1], Alfredo Ferro[3], Alfredo Pulvirenti[3]*, Vincenza Barresi[2], Salvatore Alaimo[3]

1 Institute for Biomedical Research and Innovation (IRIB), National Research Council of Italy (CNR), Palermo, Italy, 2 Department of Biomedical and Biotechnological Sciences, Section of Medical Biochemistry, University of Catania, Catania, Italy, 3 Department of Clinical and Experimental Medicine, Bioinformatics Unit, University of Catania, Catania, Italy

☯ These authors contributed equally to this work.
* alfredo.pulvirenti@unict.it

**Data Availability Statement:** The gene expression and methylation data of tumor samples and normal mucosa for COAD, BLCA, BRCA, CESC and STAD

## Abstract

Multi-layer Complex networks are commonly used for modeling and analysing biological entities. This paper presents the advantage of using COMBO (Combining Multi Bio Omics) to suggest a new role of the chromosomal aberration as a cancer driver factor. Exploiting the heterogeneous multi-layer networks, COMBO integrates gene expression and DNA-methylation data in order to identify complex bilateral relationships between transcriptome and epigenome. We evaluated the multi-layer networks generated by COMBO on different TCGA cancer datasets (COAD, BLCA, BRCA, CESC, STAD) focusing on the effect of a specific chromosomal numerical aberration, broad gain in chromosome 20, on different cancer histotypes. In addition, the effect of chromosome 8q amplification was tested in the same TCGA cancer dataset. The results demonstrate the ability of COMBO to identify the chromosome 20 amplification cancer driver force in the different TCGA Pan Cancer project datasets.

## 1. Introduction

Understanding tumor phenotype is a challenging task. Indeed, characterizing the entire tumor environment biology requires the investigation of the interactions between the different elements (i.e., genes, transcripts and proteins). In the last years, the techniques developed for generation of omics data have advanced in terms of resolution resulting in the generation of a vast amount of data using standardized methodologies [1]. On one hand, big data provides an unprecedented opportunity in biology, on the other hand great challenges arise in terms of data mining, data integration and knowledge discovery [2]. Exploring the several relations interceding in a cancer cell (environment) is crucial to stratify case subtypes or distinguish cases from controls [3]. The usage of Complex Networks to describe natural phenomena such as biological systems is increasingly common and beneficial in terms of communicative and emergent components of the system and has taken over the field [4]. This approach has been demonstrated to help reveal unknown mechanisms behind the different diseases. Indeed, the

were obtained from The Cancer Genome 86 Atlas (TCGA; https://www.cancer.gov/tcga) database.

**Funding:** AP, SA, AF, have been partially supported by the following research project: PO-FESR Sicilia 2014-2020 "DiOncoGen: Innovative diagnostics" (CUP G89J18000700007). AP, has been also partially supported by the following research project: "PROMOTE: Identificazione di nuovi biomarcatori per la diagnosi precoce di mesotelioma maligno pleurico in soggetti ex esposti a fibre asbestiformi", University of Catania - Piano di incentivi per la ricerca 2020-2022. The funders had no role in study design, data collection and analysis, decision to publish, or preparation of the manuscript.

**Competing interests:** The authors have declared that no competing interests exist.

interdependence between multiple biological elements is responsible for disease onset and progression [5, 6]. Here, the network represents a practical approach [7, 8]. A method to find expression relationships between gene pairs across the whole genome and to generate Boolean implication networks was described by Sahoo et al (2008) [9]. Many publications have used this method to identify unknown gene expression connections in several pathologies [10–13]. For these reasons, we proposed a bioinformatics framework, called COMBO, that, taking advantage of the Boolean implication method, analyze both transcriptomic and epigenomic (methylation) data in specific tumor subtypes through StepMiner and BooleanNet systems [9, 14]. The transcriptomic and epigenomic (methylation) data are also used to generate heterogeneous multi-layer graphs as previously described [15], and Neo4J is exploited to query the multi-layer network with properly defined Cypher queries. Querying the multi-layer network using a custom Cypher queries allows a deeper analysis of the group studied focusing on specific nodes and their relationships with the other nodes. For example, in our previous paper [15], we ran COMBO to generate a multi-layer network from lung adenocarcinoma (LUAD) samples having high IWS1 expression and we compared it with LUAD samples with low IWS1 expression. The multi-layers were queried in order to identify the main pathways in which IWS1 takes part. In the present work we exploit such method to tackle the more difficult task to characterize, qualitatively and quantitatively, the impact of the multigene transcriptional dysregulation induced by a specific chromosomal numerical aberration in different cancer histotypes. It is well known that large chromosomal aberrations can exert their cancer driver effects through modification of single genes at break-point regions because of the generation of gene fusion or enhancer-hijacking processes [16]. However, several numerical aneuploidies in cancer do not show any single gene structural alterations and their effects are probably mediated by cumulative gene copy-number dependent transcriptional effects [17–19]. Moreover, it is not yet clear whether some specific chromosomal aneuploidies contribute to cancer progression, exerting a significant cancer driving force, or are simply a consequence of the general chromosomal instability [20]. An interesting case is represented by trisomy or tetrasomy of specific chromosomes. Although such aberrations can be frequently observed in several types of tumors, their frequency and their time of appearance during malignant progression can largely vary across different types of tumors [21]. In the present study we focused on the increased copy number of chromosome 20 (called GAIN-Chr20) and chromosome 8q (GAIN-Chr8q), two common form of aneuploidy observed in several cancers. For each cancer type, samples bearing "GAIN-Chr20" or "GAIN-Chr8q", have been compared to cancer samples without that specific chromosomal alteration, identified as "DIS (disomic)-Chr20" or DIS-Chr8q. Therefore, we report an analysis of Boolean implication networks in tumors bearing the same chromosomal aberration in the context of the following cancer phenotypes: colon adenocarcinomas (COAD), stomach adenocarcinoma (STAD), bladder urothelial carcinoma (BLCA), breast invasive carcinoma (BRCA), and cervical squamous cell carcinoma (CESC). By a comparison of GAIN vs DIS Boolean implication networks we found that the number of nodes present in GAIN networks but absent in DIS network (GAIN differential nodes) largely varies according to chromosome and cancer type. We propose that such a method could provide an estimate of the cancer driving force of chromosome-specific numerical aberrations in the context of different cancer histotypes [15].

## 2. Materials and methods

### 2.1 Pipeline design

The pipeline consists of five phases (Fig 1). First, COMBO takes as input both expression and methylation matrices containing TPM (transcript per millions) and β-values, respectively,

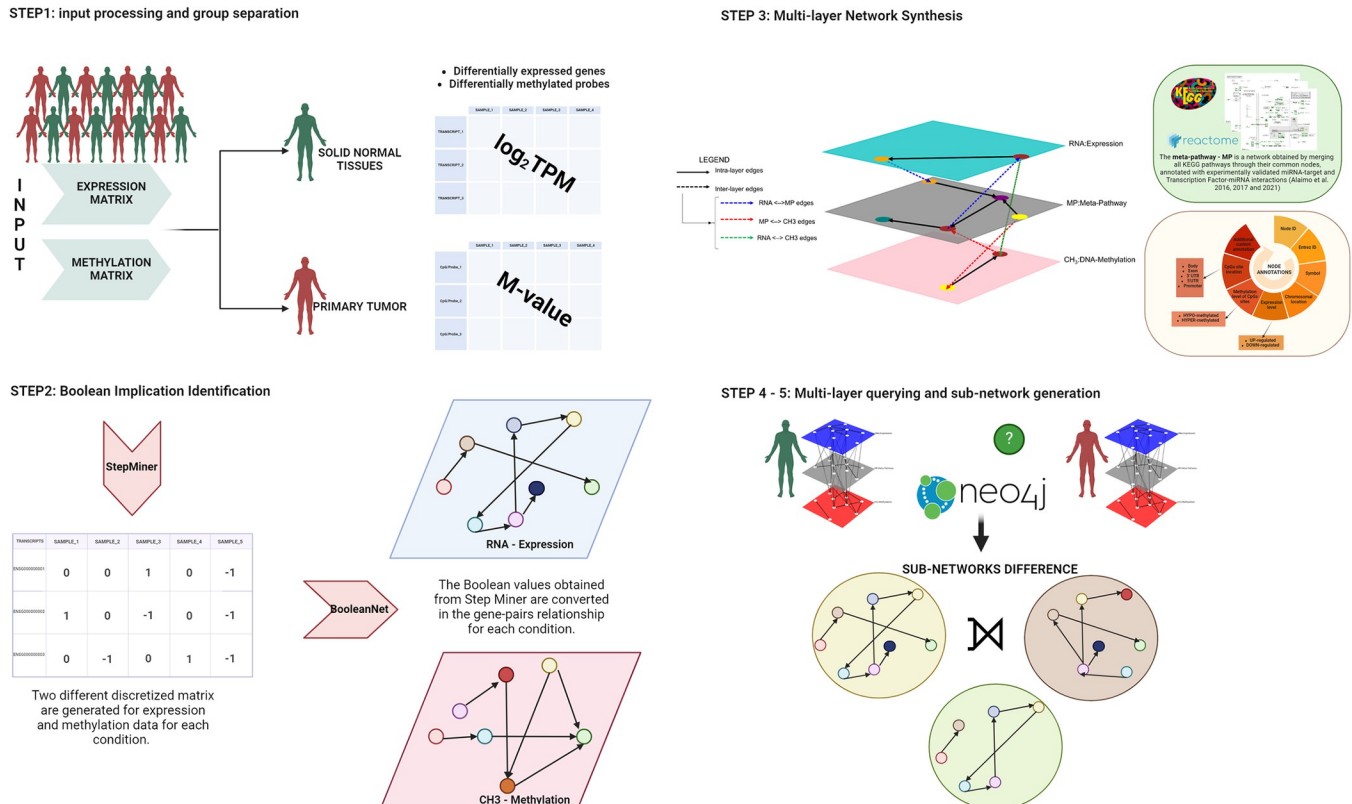

**Fig 1. The COMBO pipeline workflow.** STEP 1: TPM values in the expression matrix are log-transformed and β-value in the methylation matrix are converted in M-value. Differentially expressed gene and differentially methylated site analysis are computed. STEP 2: Boolean Implication identification for expression and methylation data. STEP 3: Multi-layer network generation for each condition under investigation. The multi-layer includes three different layers: mRNA-Expression, MP-metapathway and CH$_3$-Methylation. STEP 4–5: multi-layer interrogation using custom Cypher queries and resulting sub-network analysis. The figure was generated using BioRender.com.

along with an info table with sample information and a comparison table. The initial matrices include the unique sample identifiers in the column and the transcripts or CpG sites in the rows for expression and methylation matrices, respectively. The info table reports the list of unique identifiers for each sample and the group belonging to. COMBO exploits this table to split the input matrices for each condition. The comparison table is structured as the model matrix used in the limma R package [22]. It is used in order to identify the conditions to compare. The number of columns in the comparison table equals the number of multi-layer networks generated in the final step. In addition, an optional input table reporting custom node annotations which will be used in the querying step. These tables must be formatted with ENSEMBL ID in the rows and the group belonging to columns. A detailed explanation of the input file is reported in the S1 Appendix. Both TPM and β-values in the expression and methylation matrices are converted in log$_2$TPM and M-value, respectively (STEP 1). These will be provided as input for StepMiner [14] and BooleanNet [9] software which identify gene-pairs relationships (STEP 2). The boolean implications between the expression and methylation levels of genes are extracted in the form of "if-then" relationships. All the potential pairs of values of genes in each sample are plotted in a scatter plot which is divided, based on the StepMiner threshold, into four quadrants "low-low", "low-high","high-low", "high-high". For each pair of genes the algorithm checks which quadrant is significantly sparsely populated with points compared to the other quadrants. In the scatter plot each point represents the expression value of the pair of genes for a specific sample. Based on the empty quadrant in the scatter plot, the

algorithm detects six types of Boolean implications. The Boolean relationships can be symmetric or asymmetric. Symmetric relationships include the equivalent type, which corresponds to highly positively correlated genes, or the opposite type, which corresponds to the highly negatively correlated genes. The asymmetric ones correspond to every sparse quadrant and are divided into low—low, high—low, low—high, and high—high. In STEP 3, COMBO generates multi-layer networks for each condition under investigation. The multi-layer includes three layers (Fig 2): mRNA-Expression, MP-Metapathway, and $CH_3$-Methylation. The expression level contains nodes and directed intra-layer edges selected from BooleanNet [9] analysis conducted with RNA-seq data. Similarly, the nodes and directed intra-layer edges that participate in the $CH_3$-Methylation level are selected from BooleanNet methylation output. The MP-Metapathway layer was added to connect mRNA-expression and $CH_3$-Methylation layers. The meta-pathway is a network obtained by merging all KEGG pathways through their common nodes, annotated with experimentally validated miRNA-target and Transcription Factor-miRNA interactions [23–25]. Inter-layer edges (bidirectional) are inserted between two nodes representing the same gene to connect the other layers. Each node includes a list of annotations that the user will employ for querying the network. Finally, the multi-layer networks generated are stored in Neo4j (STEP 4–5). Then, through custom Cypher queries, the multi-layer is interrogated to generate sub-networks that can be analyzed differently (i.e., enrichment pathway analysis). The COMBO pipeline is described in more detail in Cosentini et. al, (2023) [15]. COMBO strength is in its versatility: many aspects and research areas within the field (i.e., different cancer subtypes or general pathologies) can be evaluated and compared in order to discriminate pathology specific elements. In addition, the integration of different omics data helps to identify complex interactions, patterns and correlations. Finally, the innovative querying step of COMBO reduces the dimensionality of the dataset, selecting the essential elements. Despite the advantages in using COMBO, it is relatively slow because Boolean Implications identification is a time consuming task.

## 2.2 Case study

The study reported in this paper focuses on the evaluation of increased copy number gain of whole chromosome 20 (GAIN-Chr20) or chromosome 8q (GAIN-Chr8q) as cancer driver

A.

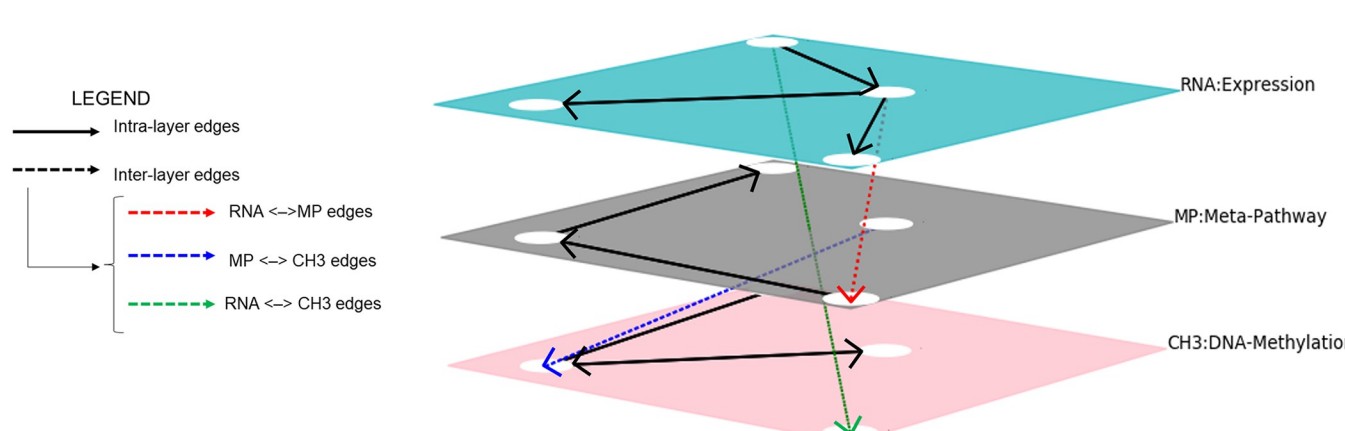

**Fig 2. A heterogeneous multi-layer network example (generated with the pymnet python library http://www.mkivela.com/pymnet/).** The intra-layer edges connect nodes belonging to the same layer. The inter-layer edges are depicted in different colors: blue indicates the cross-link between mRNA-Expression and MP-Metapathway layers; in red are shown the edges between MP-Metapathway and $CH_3$-Methylation layer; green is used to designate the connection between mRNA-Expression layer and $CH_3$-Methylation layer.

**Table 1. TCGA sample size used for COMBO evaluation.**

| DATASET | DIS-Chr20 | GAIN-Chr20 | DIS-Chr8q | GAIN-Chr8q | NORMAL TISSUE |
|---|---|---|---|---|---|
| COAD—Colon Adenocarcinoma | 67 | 85 | 109 | 117 | 19 |
| STAD—Stomach Adenocarcinoma | 117 | 146 | 128 | 143 | 63 |
| BLCA—Bladder Urothelial Carcinoma | 145 | 127 | 146 | 175 | 17 |
| BRCA—Breast Carcinoma | 370 | 146 | 299 | 282 | 84 |
| CESC—Cervical Squamous Cell Carcinoma | 127 | 55 | 136 | 60 | 3 |

aberrations using the COMBO pipeline. The gene expression and methylation data of cancer samples and normal tissues for COAD, STAD, BLCA, BRCA, and CESC were obtained from The Cancer Genome Atlas (TCGA; https://www.cancer.gov/tcga) database. The samples were grouped into three classes, cancers bearing the chromosome copy number gain (GAIN-Chr20 or GAIN-Chr8q), cancers not bearing any copy number abnormalities of that chromosome (DIS-Chr20 or DIS-Chr8q), and the corresponding normal tissue. We selected all the available projects with a statistically adequate number of samples in both GAIN-Chr20 and DIS-Chr20 groups. The corresponding clinical annotations and information for all chromosomes were obtained from the "cBioPortal for Cancer Genomics" (https://www.cbioportal.org/). The number of samples having both the RNA-seq and methylation data are reported in Table 1. The S1 File reports the list of the TCGA samples id used for the analysis. The TPM from RNA-seq and the β-values from methylation analysis were downloaded and pre-processed using the TCGAbiolinks [26] package. The statistical tests in BooleanNet are used to validate the relationship between two genes. We selected an "s threshold" that could be adapted for each TCGA dataset according to sample number and transcripts/CpGs available. BooleanNet was always used with a "s threshold" = 3 for microarray data with a relatively high number of samples [14, 27, 28]. In this study we used BooleanNet for the first time for methylation arrays. Since the methylation matrix includes about 450k probes, the s statistic is wider influenced by the higher features number. For these reasons, we selected the first available "s threshold" able to identify a reasonable number of implications for each project. Consequently, the statistical parameter used for the Boolean analysis was $s = 2$ for expression data in both chromosome 20 and 8q copy number gain. For methylation analysis we used an $s$ Threshold = 4 for chromosome 20 amplification and $s = 5$ for chromosome 8q gain. Exception for CESC GAIN-Chr20 condition, where implication with $s = 3$ were selected for methylation data. Since the number of samples in both CESC GAIN-Chr20 and GAIN-Chr8q groups is very low, no implications using $s = 4/5$ were retrieved. The "$s$ thresholds" used for BooleanNet analysis are reported in Table 2. A p-value cutoff equals to 0.01 was used to select the statistically significant Boolean Implications. The default values were chosen for the remaining parameters. False discovery rate was computed for $s = 2,3,4$ statistical threshold (FDR < 0.001) by using 20 different randomly permuting gene expression dataset as reported in [29]. The results are reported in the

**Table 2. BooleanNet "s threshold" selected for each TCGA project.**

| DATASET | s threshold Chr20 | | s threshold Chr8q | |
|---|---|---|---|---|
| | Expression | Methylation | Expression | Methylation |
| COAD—Colon Adenocarcinoma | 2 | 4 | 2 | 5 |
| STAD—Stomach Adenocarcinoma | 2 | 4 | 2 | 5 |
| BLCA—Bladder Urothelial Carcinoma | 2 | 4 | 2 | 5 |
| BRCA—Breast Carcinoma | 2 | 4 | 2 | 5 |
| CESC—Cervical Squamous Cell Carcinoma | 2 | 3 | 2 | 3 |

S2 File. For each node, we added some custom attributes to query the network. For example, we added the results of the differentially expressed genes analysis of GAIN-Chr20 vs DIS-Chr20. The genes are indicated as "OVER_T" or "DOWN_T." In addition, the protein expression, mutation status, and the transcription factor annotation for each node were also inserted. The protein expression data were retrieved from The Cancer Protein Atlas—TCPA (https://tcpaportal.org/tcpa/) for the selected samples. The expression was determined by calculating the unpaired two-sample t-test and the mean between the sample in the same group. The mutation data were collected in "cBioPortal for Cancer Genomics" for each condition. The complete list of transcription factors was retrieved from the Human Transcription Factor database (http://humantfs.ccbr.utoronto.ca/) [30]. The insertion of custom information allowed us to query the multi-layer networks more comprehensively. The multi-layer networks were interrogated using Neo4J (https://neo4j.com/). We searched for all the paths that connect the expression layer to the methylation layer including only nodes located on chromosome 20. The resulting queries, from the two conditions, were compared to maintain only the exclusive nodes and edges. Then, the remaining nodes were subjected to Reactome enrichment analysis using the ReactomePA R package [31]. Briefly, the enrichment analysis was conducted separately for each condition, selecting a p-value cutoff equals to 0.05 with the Benjamini-Hochberg correction. Then the enriched resulting data frames were combined, removing the common genes in order to generate a final image including only the genes contained in the GAIN-Chr20 network.

## 2.3 Data analysis

We introduce the term "differential node" which represents a node included in the GAIN-Chr20 or GAIN-Chr8q multi-layer network but not in the corresponding DIS-Chr20 or DIS-Chr8q multi-layer network (differential node in GAIN-Chr20/Chr8q group) or vice-versa (differential node in DIS-Chr20/Chr8q group). The percentage distribution of the differential nodes in each chromosome was evaluated in order to identify a correlation between the genes located on chromosome 20 and their implications in the analyzed phenotype. The same analysis was conducted for the genes located on the long arm of chromosome 8. We evaluated the total number of differential nodes along with the number of differential nodes located in Chr20 or Chr8q. The differential nodes in expression and methylation layer, respectively, were normalized for the total number of nodes included in the respective layer. The percentage of differential nodes in GAIN-Chr20/Chr8q was obtained by the ratio between the sum of differential nodes in the specific layer (mRNA-Expression layer or $CH_3$-Methylation) located in all the chromosomes, and the total number of nodes located in the layer analyzed. The same equation was used to compute the percentage of the differential CpG sites. 7 different analyses were conducted in order to evaluate the distribution of the totality of CpGs sites, and then the specific CpG sites: TSS200 (refers to 200–0 bases upstream to transcription start site, TSS), TSS1500 (refers to 1500–200 bases upstream TSS), 5'UTR (defined the region between the TSS and the ATG site, UnTranslated Region), Body (region between ATG and stop codons), 1st Exon and 3'UTR (defined the region between stop codon and the poly-A tail). In such a case the percentage is computed dividing the number of sites in the methylation layer representing the CpGs by the total number of CpGs sites in the $CH_3$-Methylation layer for each TCGA project. In addition, we evaluated the percentage differential nodes (genes or CpGs sites) located on the *n-th* chromosome (Chr20 or Chr8q) with respect to the total number of nodes or CpGs in the mRNA-Expression or $CH_3$-Methylation layer, respectively. The Supplementary File 3 reports all the values.

## 3. Results

### 3.1 Boolean implication networks of transcriptomic data

The COAD, STAD, BLCA, BRCA and CESC samples were retrieved from the TCGA PanCancer Atlas project, distinguishing samples having the specific chromosome numerical aberration (namely "GAIN-Chr20" and "GAIN-Chr8q) and those without such aberration (called "DIS-Chr20" and "DIS-Chr8q"). COMBO was used to generate the multi-layer networks for each condition in order to perform a comparative analysis.

Fig 3A and Table 3 report the percentage of the differential nodes selected by Boolean implication and included in the expression layer for both GAIN-Chr20 and GAIN-Chr8q (in the mRNA-Expression network nodes represent genes; see definition in section 2.3). The percentage of differential nodes in GAIN-Chr20 or GAIN-Chr8q networks across the different TCGA projects are compared. Two main conclusions can be derived: 1) the percentage of differential nodes in GAIN-Chr20 network is higher in COAD (16.93%) followed by BRCA samples (9.22%); 2) the percentage of differential nodes (genes) is higher in GAIN-Chr20 network compared to GAIN-Chr8q network in all the analyzed TCGA projects. The largest difference is observed in COAD samples (16.9% vs 2.95%). This aspect is confirmed by the percentage of differential nodes (genes) located on Chr20 with respect to total nodes COAD GAIN-Chr20 expression layer (Fig 3B).

### 3.2 Boolean implication networks of methylation data

Similar analysis has been conducted focusing on the $CH_3$-Methylation network layer in order to identify the correlation between the Chr20 or Chr8q aneuploidy and the differential nodes selected by Boolean implication. In this case all the CpG sites (Body, 1stExon, TSS200, TSS1500, 5'UTR, 3'UTR) belonging to the same gene are included in the same node (gene). Therefore, the number of differential nodes included in the $CH_3$-Methylation layer, without distinguishing which or how many CpG sites (located in that gene) concur to generate a Boolean implication, was used. The results are reported in Fig 4. The percentage reported in the graph was computed selecting only the nodes (which represent genes) included in the $CH_3$-Methylation layer. The percentage of differential nodes on the $CH_3$-Methylation network layer is reported in Fig 4. COAD, STAD and CESC show a higher percentage of differential nodes located on the $CH_3$-Methylation layer in GAIN-Chr20 (84.02%, 57.58% and 43.51% respectively) network compared to GAIN-Chr8q (33.83%,40.65% and 1.33% respectively). On the other hand, BLCA GAIN-Chr8q network includes a remarkable number of differential nodes (45.82%) with respect to GAIN-Chr20 (13.85%). No substantial differences were identified in the BRCA project, in which the percentage of differential nodes is 19.87% in GAIN-Chr20 and 17.24% in GAIN-Chr8q.

In order to evaluate how the amplification of Chr20/Chr8q affects the epigenome, we performed a more accurate analysis investigating the chromosomal percentage distribution of the differential CpGs. The differential CpG probes involved in Boolean implications were treated as in the previous analysis for the differential nodes in mRNA-Expression layer. Therefore, in this instance, a node represents a specific CpG site. Consequently, we first evaluated the differential number of total CpG sites included in the $CH_3$-Methylation layer. The percentage of the total differential CpG sites (GAIN-Chr20 vs GAIN-Chr8q) is shown in Fig 5. The results confirm the data reported for the differential nodes in the $CH_3$-Methylation layer: there is an abundance of unique CpG sites in GAIN-Chr20 for the COAD (94.17% vs 46.48%) and STAD (70.22% vs 59.08%) projects. Whereas, the difference of differential CpG sites in BRCA for the two different chromosomal aberrations is more emphasized than previous analysis, showing a

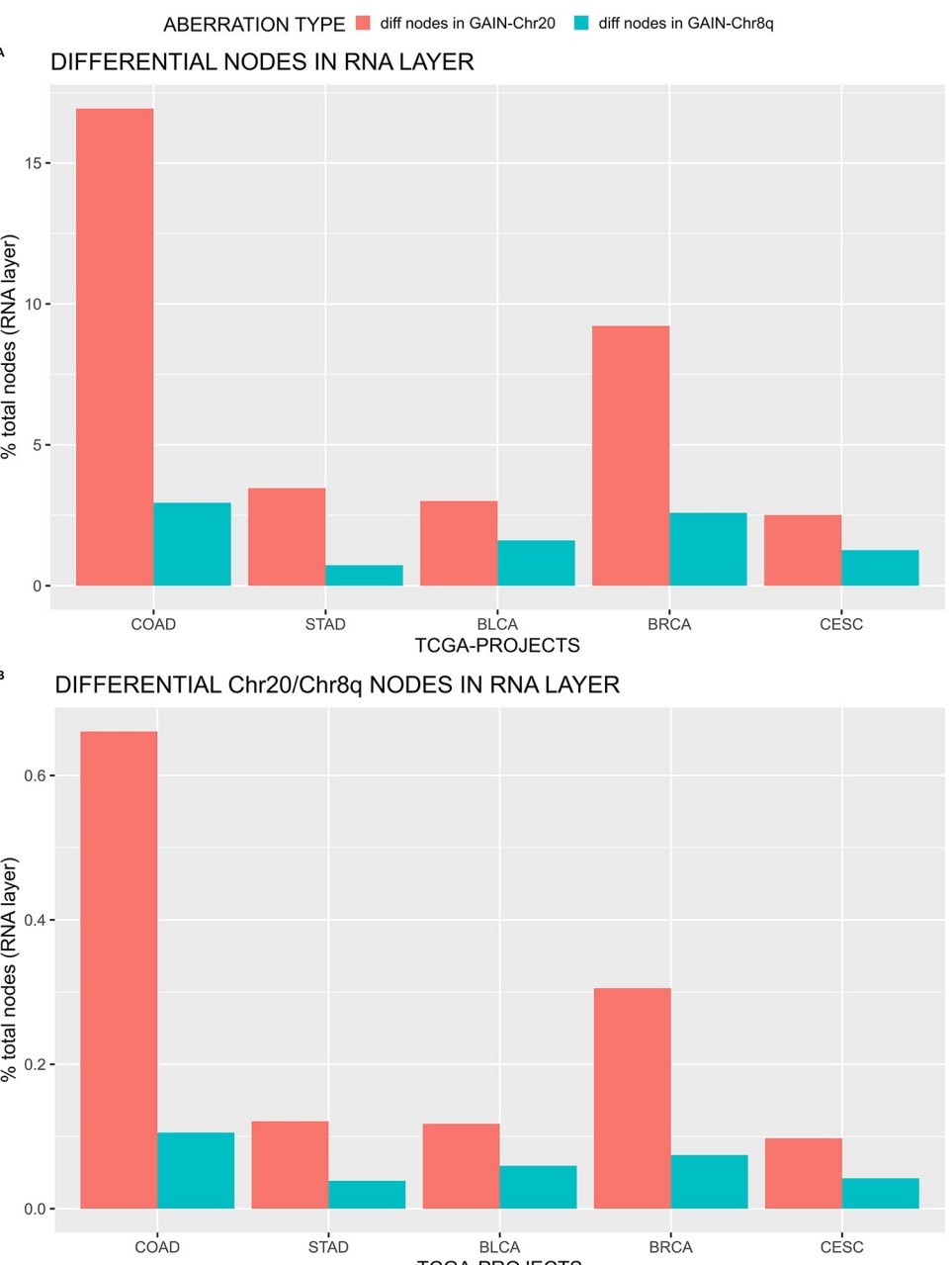

**Fig 3.** (A) Differential nodes (genes) expressed as percentage of total nodes in mRNA-Expression network layer for GAIN-Chr20 (pink) and GAIN-Chr8q (in azure) for all the analyzed TCGA projects. (B) Differential nodes (genes) located on Chr20 expressed as percentage of total nodes in mRNA-Expression network layer for GAIN-Chr20 (pink) or GAIN-Chr8q (in azure) for all the analyzed TCGA projects.

percentage of 86.06% in GAIN-Chr8q against a percentage of 22.93% in GAIN-Chr20. An opposite trend is described for BRCA and CESC, in which the percentage of differential CpG sites is higher in GAIN-Chr8q network. 40.53% and 67.86% are the percentages of all the CpG sites in GAIN-Chr8q for BRCA and CESC, respectively. These values are compared with 31.43% and 53.66% in BRCA and CESC GAIN-Chr20q networks. Then, we performed separate analysis for CpG sites located on Body, 1st Exon, TSS200, TSS1500, 5'UTR and 3'UTR.

**Table 3. Frequency of the chromosomal aberrations in TCGA samples and percentage of differential nodes in GAIN-Chr20 and GAIN-Chr8q in the analyzed TCGA projects.** The chromosomal aberration frequency was computed using the arm level copy number alteration info retrieved by "cBioPortal for Cancer Genomics" for each TCGA project.

| TCGA PROJECT | mRNA-Expression network layer GAIN-Chr20 | | mRNA-Expression network layer DIS-Chr20 | |
| --- | --- | --- | --- | --- |
| | ABERRATION FREQUENCY IN TCGA | % differential nodes | FREQUENCY IN TCGA | % differential nodes |
| COAD | 36.7% | 16,93 | 25.08% | 4,42 |
| STAD | 38.86% | 3,46 | 33.41% | 2,01 |
| BLCA | 31.14% | 3,01 | 36.74% | 2,54 |
| BRCA | 19.93% | 9,22 | 45.76% | 2,69 |
| CESC | 21.89% | 2,51 | 52.53% | 17,26 |
| | mRNA-Expression network layer GAIN-Chr8q | | mRNA-Expression network layer DIS-Chr8q | |
| COAD | 45.45% | 2,95 | 43.94% | 1,74 |
| STAD | 40.45% | 0,73 | 33.41% | 2,34 |
| BLCA | 44.04% | 1,61 | 36.25% | 2,49 |
| BRCA | 39.11% | 2,59 | 36.62% | 6,75 |
| CESC | 22.90% | 1,27 | 58.59% | 9,83 |

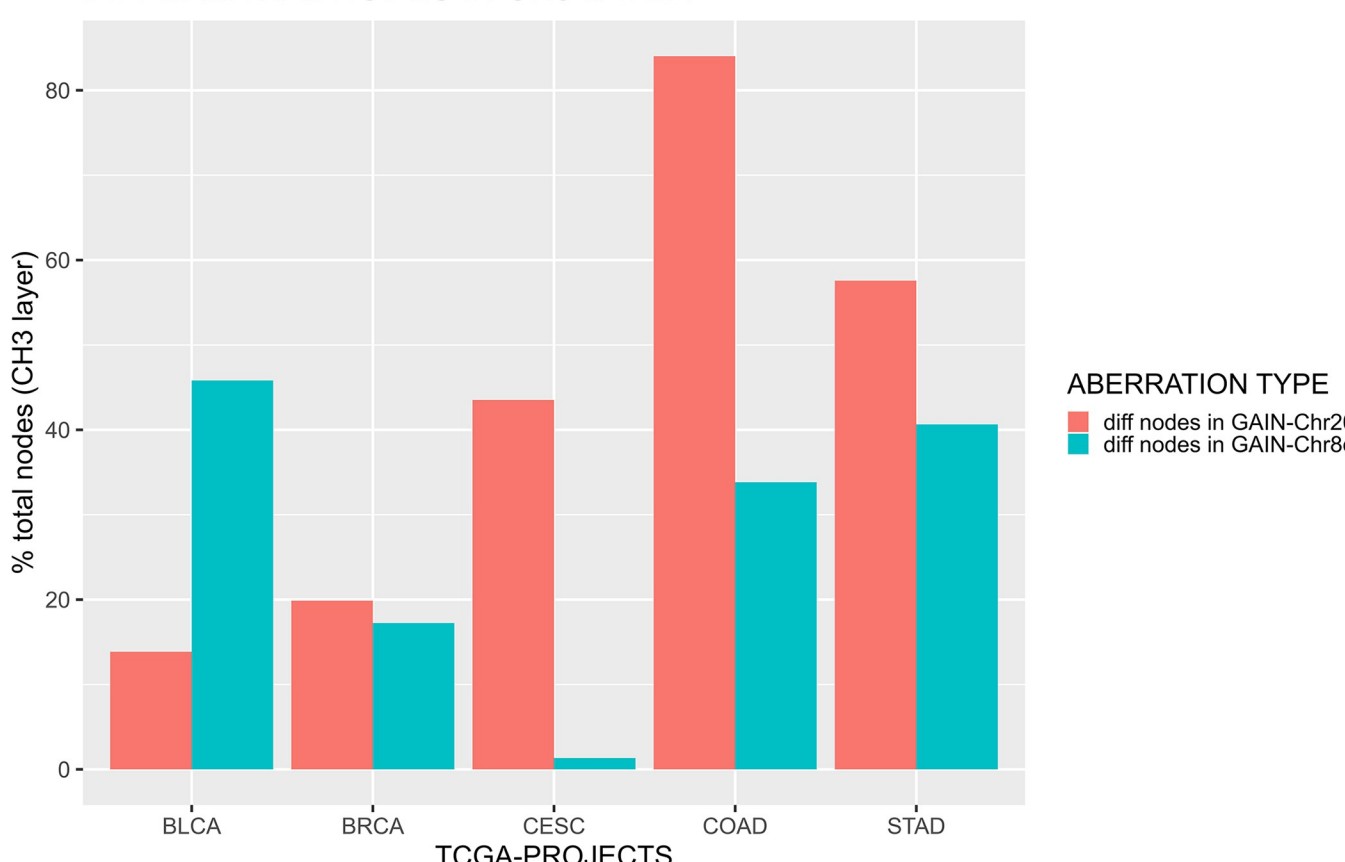

**Fig 4. Percentage of differential nodes (genes) in CH$_3$-Methylation network across the analyzed dataset for both GAIN-Chr20 and GAIN-Chr8q networks.** The percentage was computed taking into consideration a node in the methylation layer as the combination of the total CpG sites (Body, 1stExon, TSS200, TSS1500, 5'UTR, 3'UTR) located in that gene.

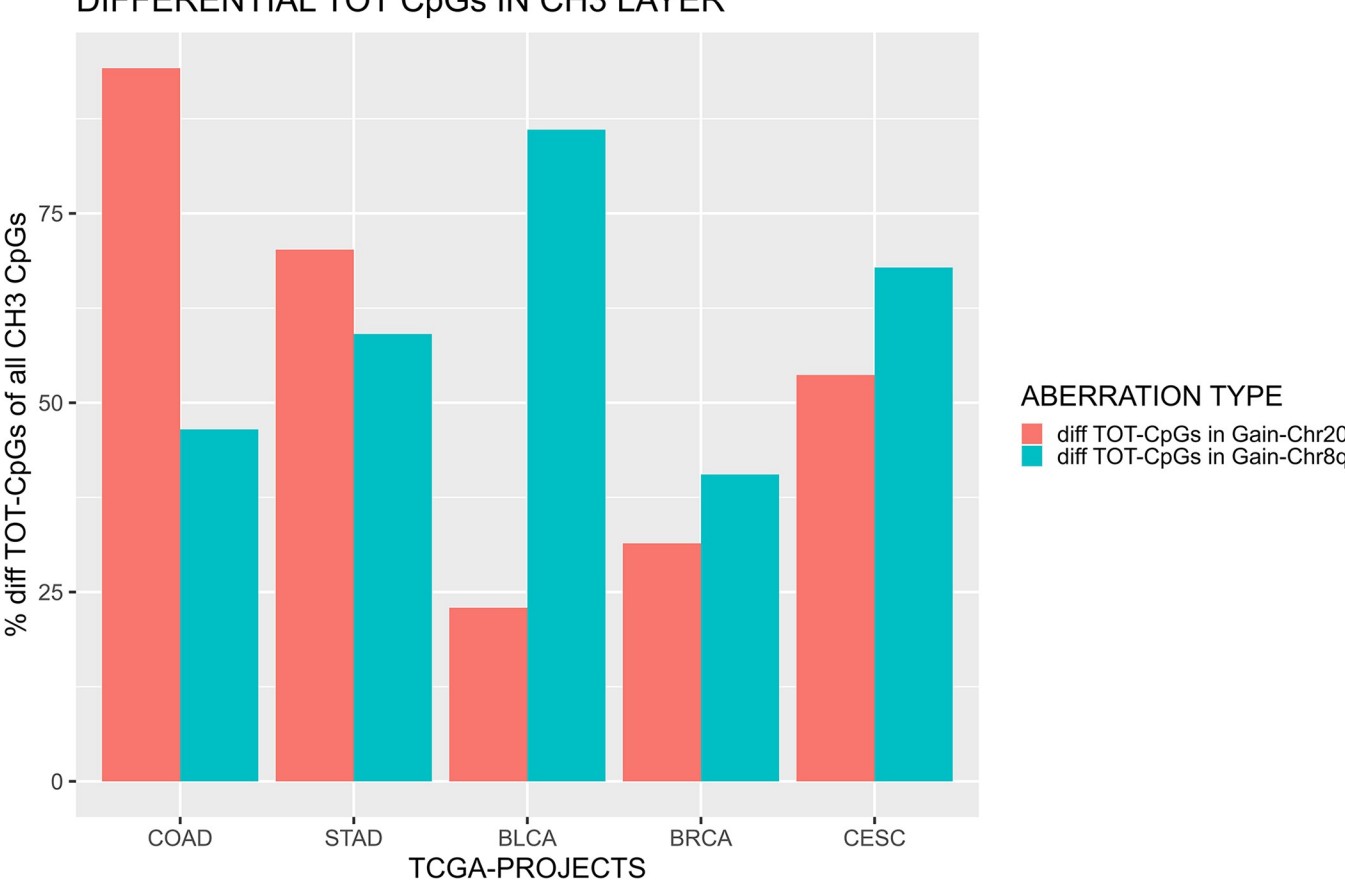

**Fig 5. Differential CpG sites in the analyzed dataset.** The percentage was computed taking into account a node in the methylation layer as a specific CpG site of a gene.

For this analysis, the percentage of differential sites was computed dividing by the total number of CpG sites in the $CH_3$-Methylation layer. Results are reported in S3 File and S1 and S2 Figs. The results confirm a higher incidence of Boolean connections in GAIN-Chr20 compared with GAIN-Chr8q for COAD across the different CpG sites. In addition, the 3'UTR is enriched in STAD GAIN-Chr20. On the other hand, the percentage of differential CpG sites is higher in GAIN-Chr8q.

### 3.3 Differential nodes in Boolean implication networks and functional pathway enrichment

A functional pathway enrichment analysis of the differential nodes (GAIN vs DIS) in both expression and methylation layers was performed. Fig 6 shows the top 10 enriched pathways in each TCGA project subdivided for expression (mRNA) and methylation ($CH_3$) layers. BLCA and CESC results are not reported in the figure because of the low number (<3) of significant enriched pathways. The color indicates the adjusted p-value scale (cutoff = 0.05), whereas the size represents the percentage of significant genes over the total genes in a given pathway (gene ratio). Only the "NCAM1: interaction" pathway is common between the different tumors analyzed. The selection of the top 50 pathways confirmed the absence of intersections between the different TCGA dataset. The enriched pathways are mainly composed of

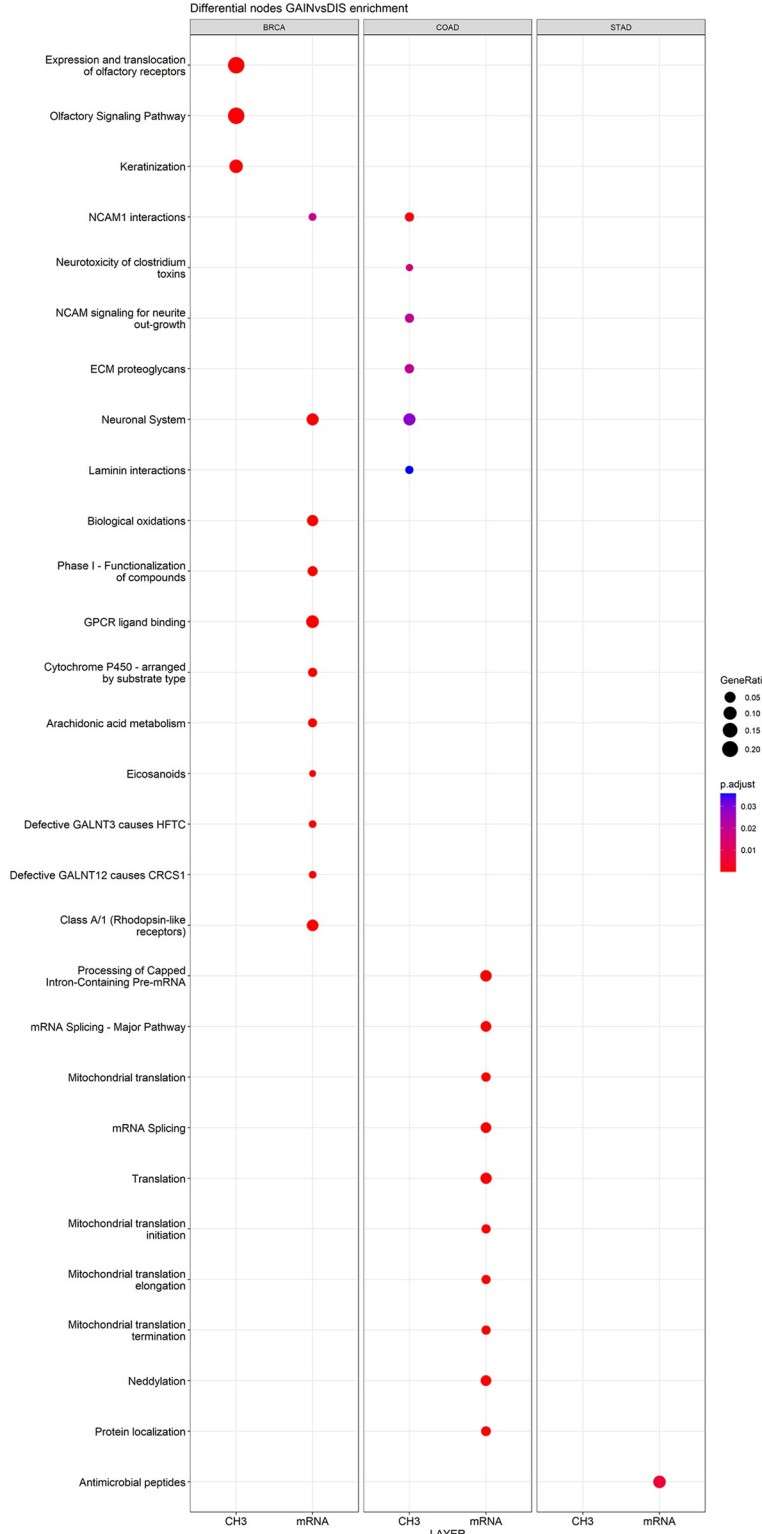

**Fig 6. Dotplot showing the top 10 enriched pathways using differential nodes in GAIN-Chr20 groups for COAD, STAD and BRCA.** The x-axis shows the enriched pathways (p-value = 0.05). BLCA and CESC results are not significant and consequently excluded from the plot. The results are reported for each project separating differential nodes in GAIN-Chr20 belonging to expression (mRNA) and methylation (CH$_3$) layers (as indicated in y-axis). The dot color indicates the adjusted p-value as listed in the color bar in order to notify the significance of the enriched term. The dot size represents the gene ratio.

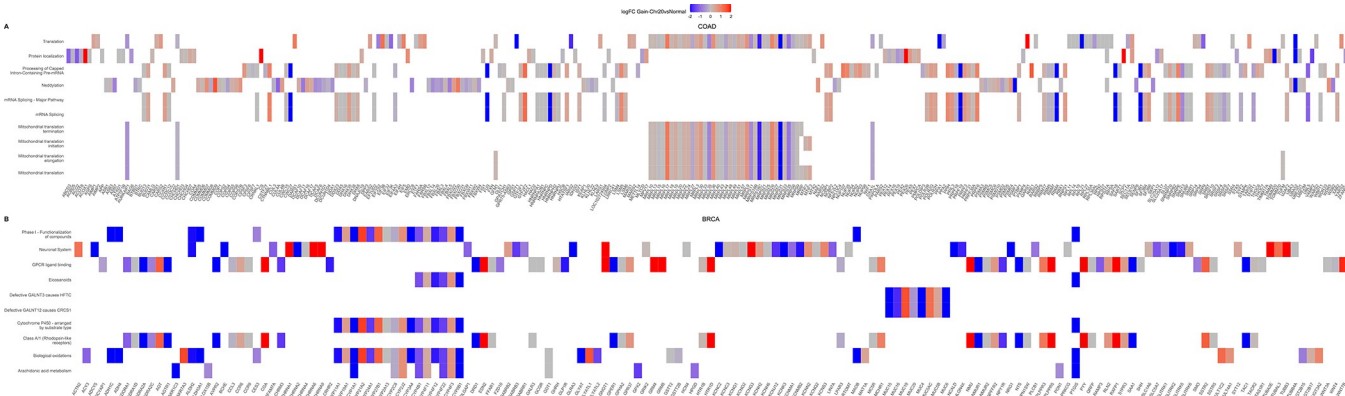

**Fig 7.** Heatplots of the Reactome enriched pathways for the (A) COAD, (B) BRCA. The plots were generated using the differential nodes in GAIN-Chr20 group located in the mRNA-expression layer. The y-axis reports the significant enriched pathways. The x-axis indicates the genes contributing to the pathway. In gray are indicated the genes with logFC = 0. The color key represents a spectrum of lowest gene expression (blue) to highest gene expression (red). In order to reduce the color scale, we setted a logFC range between 2 and -2.

differential expression layer nodes in GAIN-Chr20 groups. Indeed, the unique nodes in the methylation layers do not concur significantly to specific pathways. Consequently, we focused on the expression layer and we studied the genes involved in the enriched pathways. The logFC comparing GAIN-Chr20 to Normal Tissues group was reported for each gene. A total of 183 significant Reactome terms (p-value cutoff = 0.05) were identified in COAD, as shown in Fig 7(A). The most significantly enriched Reactome pathways were"Translation", "Protein localization", "Processing of Capped Intron-Containing Pre-mRNA" and "mRNA Splicing—Major Pathway". The significant pathways for BRCA (63 in total) were mainly enriched in the"Phase I—Functionalization of compounds", "Neuronal System" and "GPCR ligand binding", as shown in Fig 7(B). No pathways are in common between COAD and BRCA.

## 3.4 Multi-layer integration and functional pathway enrichment

The multi-layer networks, generated by COMBO for COAD, STAD, BLCA, BRCA and CESC TCGA-projects subdivided in GAIN-Chr20 group and DIS-Chr20 group, were interrogated. The Cypher custom query was generated to identify all the possible paths that connect the expression layer to the methylation layer, considering only nodes located on chromosome 20. The resulting GAIN-Chr20 and DIS-Chr20 sub-networks were compared in order to remove the common pathways. The final GAIN-Chr20 sub-network difference includes only the nodes and respective links specific to the GAIN-Chr20 subnetwork and not present in the DIS-Chr20 sub-network. A Reactome enrichment analysis of the subnetwork nodes in GAIN-Chr20 group selected using the above-mentioned Cypher query was performed for each TCGA-Project. The heat map describing the top enriched pathway with the corresponding genes belonging to that specific pathway, is reported in Fig 8. The color represents the logFC between GAIN-Chr20 and Solid Normal Tissues condition. 22 pathways are common among all analyzed datasets: COAD, STAD, BLCA, BRCA and CESC. The genes included in the sub-network difference for the COAD project involve 59 significant Reactome terms (p-value cutoff = 0.05). The Fig 8(A) shows the most significantly enriched pathways, including "Transcription of EF2 targets under negative control by p107 and p130", "TP53 Regulates transcription of genes involved in G2 cell cycle arrest", "Signal regulatory protein family interaction" and "Response of EIF2AK1 to heme deficiency". These pathways are similar across the different projects. 52 significant Reactome pathways have been detected for the STAD project

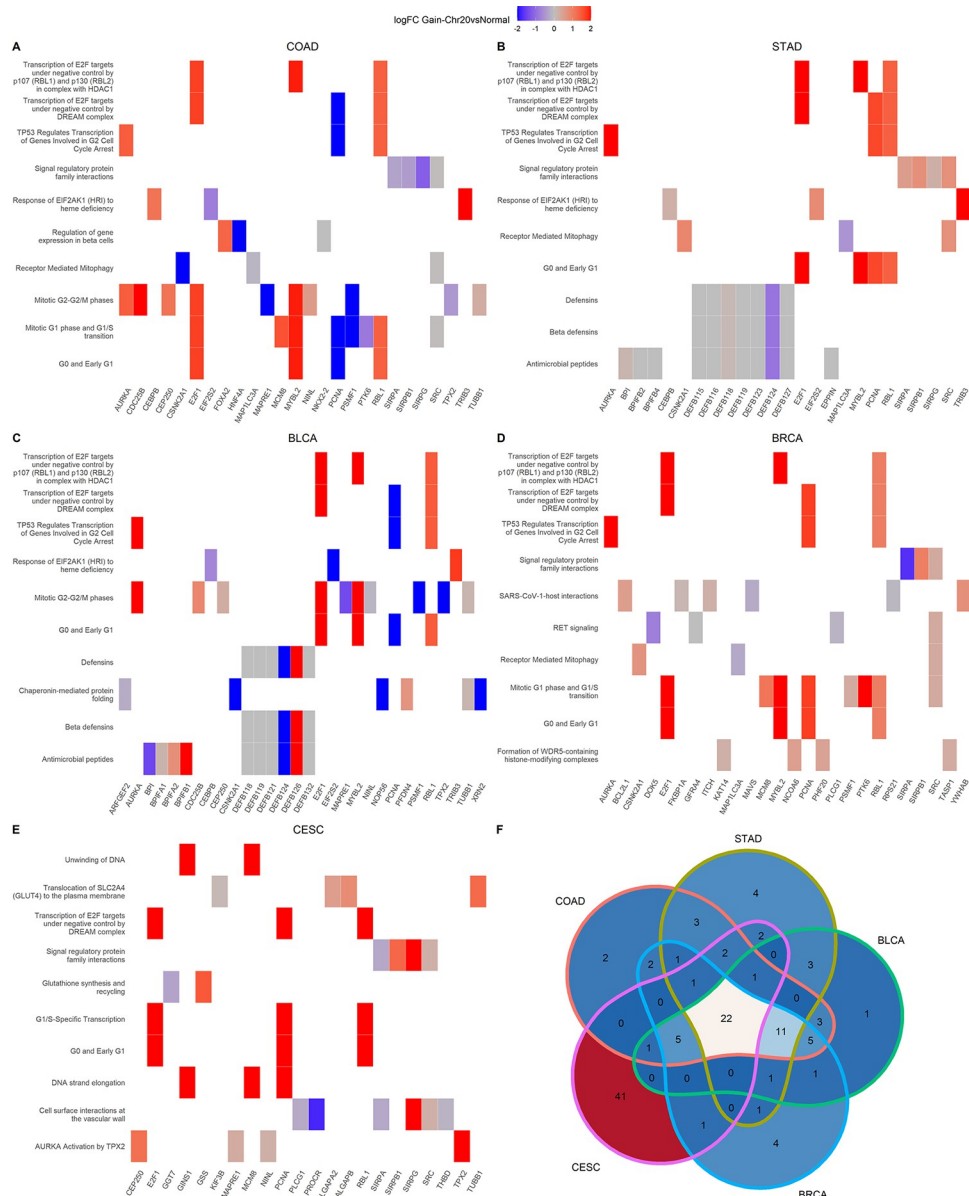

**Fig 8.** Heatplots of the Reactome enriched pathways for the (A) COAD, (B) STAD, (C) BLCA, (D)BRCA and (E) CESC projects. The plots were generated using the sub-network nodes resulting from the COMBO querying step. The enriched pathways are listed on the vertical axis. The gene symbols contributing to the pathway are reported in the horizontal axis. The gene logFC computed between GAIN-Chr20 and Solid Normal Tissue is represented using blue and red for the lowest and highest gene expression, respectively. A logFC range between 2 and -2 was setted to reduce the color scale. Gray indicates genes with logFC = 0. (F) Venn Diagram with common enriched pathways for the analyzed TCGA projects.

(Fig 8(B)), where MYBL2 and AURKA are the most upregulated genes. The total pathways in common between COAD and STAD are 41. The nodes belonging to the subnetwork difference in the BLCA project (Fig 8(C)) are enriched in 54 Reactome pathways, of which 48 are in common with COAD. MYBL2 and CSNK2A1 are the genes with the highest and lowest expression, respectively. The significant pathways in the BRCA project are 55 of which 47 for STAD and 37 for COAD are among those mainly enriched. BRCA and BLCA have 45 (81,8%) significant

enriched pathways in common. Both AURKA and MYBL2 are the genes with the highest expression (Fig 8(D)). Finally, 76 enriched pathways have been detected in the CESC project, such as "Translocation of SLC2A4 to the plasma membrane". The genes TPX2 and PROCR are respectively the most upregulated and downregulated belonging to those pathways, as shown in Fig 8(E). The Venn graph in Fig 8(F) sums up the counts of the common significant enriched pathways between the five different projects.

## 4. Discussion

COMBO is an easy-to-use bioinformatics pipeline to generate and analyze multi-layer networks combining multi-omics data. The researcher can use this method for a preliminary study to identify critical genes in a specific disease and evaluate the interplay between DNA methylation and transcription. Finally, the multi-layer approach is also helpful for the biomedical and research lab to filter the vast amount of RNA-seq and methylation data [15]. In order to demonstrate the abilities of COMBO, we analyzed different TCGA PanCancer Atlas project datasets (COAD, BLCA, BRCA, CESC and STAD) in order to study the effect of numerical chromosomal aberrations, such as GAIN-Chr20 and GAIN-Chr8q, on expression and methylation Boolean implication networks. Currently, while the strategy for distinction between cancer driver and passenger mutations is relatively well-established for single-gene or oligo-gene mutational events, the task is more complicated for broad chromosomal numerical aberrations that, through one or few chromosome copy-number dependent events simultaneously implicate several adjacent genes [32–34]. Whole-chromosome or chromosomal arm gains or losses and aneuploidy are a near-universal characteristic of human cancers that occurs in 88% of samples [21]. In Condorelli et al. 2018, 2019 the influence of BCNAs on transcriptome profile by applying SNP- and transcriptome-arrays to a series of colorectal cancer (CRC) samples bearing chromosomal instability (80%) have been investigated showing changes in expression due to highly recurrent Broad Copy Number Gains in chromosome 20, 8, 7, 13 (BCNGs in chromosomes 20, 8, 7, 13) [32, 33]. It has been suggested that the cancer driving effect of simple chromosome numerical aberrations, such as trisomy or tetrasomy of specific whole chromosome or chromosome arm, might be mediated by cumulative gene copy-number dependent transcriptional effects [12–14]. However, the analysis of simple gene expression level might not distinguish passenger effects from cancer driving ones and the analysis of classical gene expression correlation may overlook important non-linear relationships. As reported in [9], Boolean implications methods can capture more gene expression relationships and might represent a better tool for analysis of gene expression modifications induced by chromosome gain aberrations. Interestingly, our analysis of Boolean implication networks revealed a larger contribution of genes specifically present in Boolean implication networks in the case of COAD bearing GAIN-Chr20. Our result in COAD is in agreement with the observation that GAIN-Chr20 is a frequent and early-appearing abnormality in colon cancer [35–38], thus suggesting that GAIN-Chr20 exerts a stronger cancer driving force in the context of colon cancer in comparison to other cancer types, such as STAD, BLCA and CESC. Indeed, several research groups have reported the importance of Chr20 amplification in colon cancer, and attributed such effect to the location of multiple oncogenes such as BCL2L1, AURKA, TPX2, and SRC on this chromosome [39–43]. Although the involvement of those oncogenes might play a role in the cancer progression of any tumor type bearing GAIN-Chr20, their simple expression level is not able to predict the strong cancer driving force of such chromosomal aberration in colon cancer. Therefore, we propose that the proportion of genes contributing to functional networks identified by Boolean implications is a better parameter to evaluate the cancer driving force of GAIN-Chr20 aberration. Unexpectedly, BRCA samples bearing

GAIN-Chr20 also showed a relatively high proportion of specific genes (differential nodes) in gene expression Boolean implication networks. This is in contrast with the observation that GAIN-Chr20 is not very frequent in breast adenocarcinomas (see Table 2). However, frequency of aberrations is not a reliable proxy for cancer driver strength and it is possible that a subgroup of BRCA (bearing GAIN-Chr20) is strongly dependent on such aberrations. Only experimental studies aimed to modify gene expression in such selected BRCA subgroup may provide further evidence on this point. Boolean implication network analysis can provide essential guidance to such experiments. When we analyzed the list of specific genes participating in the Boolean implication networks, focusing on the mRNA-Expression layer of GAIN-Chr20 by functional enrichment analysis, the enriched pathways across the TCGA project studied did not show any overlapping. This result suggests that strong cancer driver properties of GAIN-Chr20 might be mediated by completely different multigene-cooperation patterns in the context of different cancer types. On the other hand, analyzing the nodes representing genes located on Chr20 that are linked in the multi-layer, we observed a set of genes and respective enriched pathways in common across the projects. Indeed, some genes correlated with Chr20 amplification (i.e. MYBL2, AURKA and EIF2S2) are included in all the subnetworks selected by Neo4j query. Consequently, the pathways mainly enriched are similar across the projects. Probably, the effect of GAIN-Chr20 is not circumscribed to the chromosome altered, but the aberration affects in turn different pathways and genes located in different chromosomes. For these reasons, we observed a redundancy of information selecting only genes located on Chr20, but not analyzing the whole mRNA-Expression network layer. In conclusion, we have shown that hidden information stored in gene expression Boolean implication network is able to provide a quantitative estimate of the cancer driving force of GAIN-Chr20 and such ability can be now tested in a wider range of chromosomal aberrations and cancer types. Indeed, the multi-layer-method (COMBO) could be exploited to filter the crucial elements in different chromosomal aberrations which collaborate to execute the cancer driving effect. The identification of key pathways and specific genes which specifically act in a phenotype, could improve the identification of molecular targets. Although the method is innovative, we are planning to improve some limitations of COMBO. First of all, the prediction of the cross-link between two layers is a mandatory task. In this version, COMBO inserts an inter-layer edge between two nodes in different layers that represent the same entities. These limitations can be overcome by predicting the inter-layer edge using the Boolean Implication. It involves the reduction of the inter-layer edges number and the introduction of a specific direction (no more bidirectional edges). Adopting these modifications, other omics data can be introduced in order to study the complex interplay in different pathologies. On one hand, we used BoolenNet for the first time for the imputation of connection between CpG sites. On the other hand, this step of COMBO is highly time consuming. To overcome this limitation, we aim to either enhance our method for identifying the intra-layer edges or find a better alternative. The proposed improvements will enable us to explore how different chromosomal aberrations drive cancer development.

## Supporting information

**S1 Fig.** Percentage of TSS200 (A), TSS1500 (B) and 5'UTR(C) differential CpG sites in COAD, STAD, BLCA, BRCA and CESC.
(TIF)

**S2 Fig.** Percentage of Body (A), 1st Exon (B) and 3'UTR(C) differential CpG sites in COAD, STAD, BLCA, BRCA and CESC.
(TIF)

**S1 File. TCGA samples ID list.**
(XLSX)

**S2 File. FDR results.**
(XLSX)

**S3 File. Multilayer nodes analysis in both mRNA and CH3 layer.**
(XLSX)

**S1 Appendix. COMBO input preparation and explanation.**
(DOCX)

## Author Contributions

**Conceptualization:** Ilaria Cosentini, Daniele Filippo Condorelli, Alfredo Ferro, Alfredo Pulvirenti, Vincenza Barresi, Salvatore Alaimo.

**Data curation:** Ilaria Cosentini, Daniele Filippo Condorelli, Giorgio Locicero, Alfredo Ferro, Alfredo Pulvirenti, Vincenza Barresi, Salvatore Alaimo.

**Formal analysis:** Ilaria Cosentini, Daniele Filippo Condorelli, Giorgio Locicero, Alfredo Ferro, Alfredo Pulvirenti, Vincenza Barresi, Salvatore Alaimo.

**Funding acquisition:** Daniele Filippo Condorelli, Alfredo Ferro, Alfredo Pulvirenti, Vincenza Barresi, Salvatore Alaimo.

**Investigation:** Daniele Filippo Condorelli, Alfredo Ferro, Alfredo Pulvirenti, Vincenza Barresi, Salvatore Alaimo.

**Methodology:** Ilaria Cosentini, Daniele Filippo Condorelli, Giorgio Locicero, Alfredo Ferro, Alfredo Pulvirenti, Vincenza Barresi, Salvatore Alaimo.

**Software:** Ilaria Cosentini, Giorgio Locicero.

**Supervision:** Daniele Filippo Condorelli, Alfredo Ferro, Alfredo Pulvirenti, Vincenza Barresi, Salvatore Alaimo.

**Validation:** Ilaria Cosentini, Daniele Filippo Condorelli, Alfredo Ferro, Alfredo Pulvirenti, Vincenza Barresi, Salvatore Alaimo.

**Visualization:** Ilaria Cosentini, Daniele Filippo Condorelli, Giorgio Locicero, Alfredo Ferro, Alfredo Pulvirenti, Vincenza Barresi, Salvatore Alaimo.

**Writing – original draft:** Ilaria Cosentini, Daniele Filippo Condorelli, Giorgio Locicero, Alfredo Ferro, Alfredo Pulvirenti, Vincenza Barresi, Salvatore Alaimo.

**Writing – review & editing:** Ilaria Cosentini, Daniele Filippo Condorelli, Giorgio Locicero, Alfredo Ferro, Alfredo Pulvirenti, Vincenza Barresi, Salvatore Alaimo.

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
