## [Decision Letter · Decision Letter 0]

8 Aug 2023

PONE-D-23-20487Measuring Cancer Driving Force Of Chromosomal Aberrations Through Multi-layer Boolean Implication NetworksPLOS ONE

Dear Dr. Pulvirenti,

Thank you for submitting your manuscript to PLOS ONE. After careful consideration, we feel that it has merit but does not fully meet PLOS ONE’s publication criteria as it currently stands. Therefore, we invite you to submit a revised version of the manuscript that addresses the points raised during the review process.

We look forward to receiving your revised manuscript.

Kind regards,

Chen Li, Ph.D.

Academic Editor

PLOS ONE

“AP, SA, AF, have been partially supported by the following research project: PO-FESR Sicilia 2014-2020 “DiOncoGen: Innovative diagnostics” (CUP G89J18000700007). AP, has been also partially supported by the following research project: “PROMOTE: Identificazione di nuovi biomarcatori per la diagnosi precoce di mesotelioma maligno pleurico in soggetti ex esposti a fibre asbestiformi”, University of Catania - Piano di incentivi per la ricerca 2020-2022.”

3. We notice that your supplementary figures are uploaded with the file type 'Figure'. Please amend the file type to 'Supporting Information'. Please ensure that each Supporting Information file has a legend listed in the manuscript after the references list.

Reviewers' comments:

Reviewer's Responses to Questions

**Comments to the Author**

1. Is the manuscript technically sound, and do the data support the conclusions?

Reviewer #1: Yes

Reviewer #2: Yes

Reviewer #3: Partly

2. Has the statistical analysis been performed appropriately and rigorously? 

Reviewer #1: Yes

Reviewer #2: Yes

Reviewer #3: No

3. Have the authors made all data underlying the findings in their manuscript fully available?

Reviewer #1: Yes

Reviewer #2: Yes

Reviewer #3: Yes

4. Is the manuscript presented in an intelligible fashion and written in standard English?

Reviewer #1: Yes

Reviewer #2: Yes

Reviewer #3: Yes

5. Review Comments to the Author

Reviewer #1: Materials and methods:

Pipeline design:

• It would be helpful to consider providing more information about the specific input files described in [15]. Additionally, mentioning the purpose or relevance of these input files would enhance the understanding of the pipeline's setup.

• It would be beneficial to include a brief overview or rationale of the overall objective and significance of the pipeline, suggesting how it might address existing challenges in cancer research or omics data analysis. This could help readers understand the motivation behind the pipeline and its potential impact in the field.

Case study:

• It might be useful to provide more details about the Reactome enrichment analysis using the ReactomePA R package. A brief explanation of how the enrichment analysis was conducted, including the specific settings or thresholds used to identify enriched pathways, could be suggested. Additionally, mentioning the significance level or multiple testing correction method employed in the enrichment analysis would improve the reproducibility and transparency of the case study, allowing other researchers to replicate the analysis and interpret the results more effectively.

Data analysis:

• The authors introduce the term "NCDI" (Normalized Chromosomal Index). To ensure clarity, it might be helpful to move the definition of this acronym to a location where it is more relevant to the context.

Results:

Boolean Implication networks of transcriptomic data:

• Suggest discussing the significance of results: The results show NCDI values for differential nodes in different cancer types. Including an interpretation of the implications or significance of these findings and relating them to the research questions or hypotheses being addressed would enhance the discussion section.

• Suggest providing more details on the statistical analysis: The Pearson correlation index and p-value mentioned in the text should be further elaborated upon. Providing more details about the statistical methods used for this analysis and how they interpret the correlation between NCDI ratio and the frequency of GAIN-Chr20 could enhance the readers' understanding.

Differential nodes in Boolean implication networks and functional pathway enrichment:

• Suggest including a more thorough interpretation of the results of the functional pathway enrichment analysis for different TCGA projects. Elaborating on the biological implications of the identified enriched pathways and their relevance to the research objectives or the known biology of the studied tumors with chromosome-20 gain would make the discussion more insightful.

• For Figure 6, suggest expanding the caption to describe the axes, labels, and how the dots were generated (e.g., significance, gene ratio). This additional information will help readers better understand the information conveyed in the figure without having to refer to additional materials or methods sections.

Discussion:

• Consider discussing potential limitations of the study and the COMBO pipeline to provide a balanced perspective. Mentioning any data-related or analytical constraints that might impact the interpretation of the results would strengthen the discussion.

• Suggest providing insights into potential future research directions based on the findings. Discussing how the results from this study could inform future studies or guide further investigations in cancer genomics and computational biology would add value to the discussion section.

Reviewer #2: The manuscript does an excellent job of exploring the application of the COMBO (COMBining multi-Omics) bioinformatics pipeline for the integration and analysis of multi-layer networks combining various forms of omics data. The work demonstrates the strength of COMBO in identifying critical genes in specific diseases and evaluating the interaction between DNA methylation and transcription. The authors' demonstration of COMBO's utility in analyzing multiple TCGA PanCancer Atlas project datasets effectively underscores its potential in cancer research.

However, I would like to propose a few minor suggestions for consideration:

Discussion of Results: While the authors presented a robust argument for the use of COMBO in the analysis of numerical chromosomal aberrations like GAIN-Chr20, further clarity would be beneficial. Specifically, expanding on the methodology and rationale behind the selection of these aberrations and the consequences of these aberrations on different cancers could be insightful.

Additional Use Cases: Given the demonstrated success of the COMBO pipeline in filtering vast RNA-seq and methylation data, it would be interesting to see it applied to other complex datasets. Exploring this further could provide stronger validation of the approach's versatility and robustness.

Clarification of Terminology: While the manuscript is largely well-written, some terminologies (such as Boolean implication networks) might not be familiar to all readers. I recommend including brief explanations or references for these terms to make the paper more accessible to a broader audience.

Future Work: The authors suggest that the information contained in gene expression Boolean implication networks can provide a quantitative estimate of the cance r-driving force of GAIN-Chr20. This proposition is exciting and potentially groundbreaking. It would be valuable if the authors could discuss more on how this method can be further exploited in future research, perhaps by providing some potential scenarios or implications.

In conclusion, I found the manuscript to be of high quality, presenting a compelling case for the use of COMBO in the analysis of multi-omics data and its potential impact on cancer research. These minor suggestions are intended to polish the already excellent work and help the authors in disseminating their findings to the widest possible audience.

Reviewer #3: In this work by Cosentini et al., the authors introduce a novel pipeline that helps to get a systems view of multi-omics data by analyzing heterogeneous multilayered networks constructed from transcriptomic, epigenetic, and metabolic databases. They also evaluate their pipeline (COMBO) in various TCGA cancer datasets. As a result of their analysis, they identify broad gain in chromosome 20 on different cancers as a driving force. Though their pipeline is novel and could be a useful tool in medical research, there is a lack of clarity in their pipeline which might affect the results significantly and might change the conclusion. If the authors clarify the following details and also explain the consequence of altered consideration, this pipeline could be a good tool for medical research. I recommended a revision of the work.

Specific points:

1. During the Boolean network constriction, the authors have used different values of s-statistics ranging from 2-4, which might affect the strength of the network very much.

2. The authors have not described the p-statistics cutoff they used during network construction.

3. It is not clear if they used a Boolean clustered network or simply use all the genes as separate nodes. If they used each gene as a separate node, then the size of the network and edge distribution will change the result significantly. If they use the clustered network, they haven’t described the cutoffs used in the cluster formation of different Boolean relations.

4. The terminology they have used is not consistent throughout the paper. Which could be confusing for the reader.

5. The authors haven’t described the statistical cutoff used and their logic during the identification of the interlayer edges from KEGG pathways.

6. The authors evaluate their pipeline by identifying chromosome 20 over various cancers. I believe the analysis of the gain or loss of at least another chromosome is required to properly evaluate the pipeline.

6. PLOS authors have the option to publish the peer review history of their article (what does this mean?). If published, this will include your full peer review and any attached files.

Reviewer #1: **Yes: **Haotian Zheng

Reviewer #2: No

Reviewer #3: **Yes: **Saptarshi Sinha

---

## [Author Response · Author response to Decision Letter 0]

2 Mar 2024

Please find the enclosed response_to_reviewers.pdf

---

## [Decision Letter · Decision Letter 1]

20 Mar 2024

Measuring Cancer Driving Force Of Chromosomal Aberrations Through Multi-layer Boolean Implication Networks

PONE-D-23-20487R1

Dear Dr. Alfredo Pulvirenti,

We’re pleased to inform you that your manuscript has been judged scientifically suitable for publication and will be formally accepted for publication once it meets all outstanding technical requirements.

An invoice for payment will follow shortly after the formal acceptance. To ensure an efficient process, please log into Editorial Manager at Editorial Manager® , click the 'Update My Information' link at the top of the page, and double check that your user information is up-to-date. If you have any billing related questions, please contact our Author Billing department directly at authorbilling@plos.org.

Kind regards,

Chen Li, Ph.D.

Academic Editor

PLOS ONE

Additional Editor Comments (optional):

Reviewers' comments:

Reviewer's Responses to Questions

**Comments to the Author**

1. If the authors have adequately addressed your comments raised in a previous round of review and you feel that this manuscript is now acceptable for publication, you may indicate that here to bypass the “Comments to the Author” section, enter your conflict of interest statement in the “Confidential to Editor” section, and submit your "Accept" recommendation.

Reviewer #2: All comments have been addressed

Reviewer #3: All comments have been addressed

2. Is the manuscript technically sound, and do the data support the conclusions?

Reviewer #2: (No Response)

Reviewer #3: Yes

3. Has the statistical analysis been performed appropriately and rigorously? 

Reviewer #2: Yes

Reviewer #3: Yes

4. Have the authors made all data underlying the findings in their manuscript fully available?

Reviewer #2: Yes

Reviewer #3: Yes

5. Is the manuscript presented in an intelligible fashion and written in standard English?

Reviewer #2: Yes

Reviewer #3: Yes

6. Review Comments to the Author

Reviewer #2: (No Response)

Reviewer #3: The authors provided satisfactory explanations/modifications for all the quarries in their revised manuscript.

7. PLOS authors have the option to publish the peer review history of their article (what does this mean?). If published, this will include your full peer review and any attached files.

Reviewer #2: No

Reviewer #3: **Yes: **Saptarshi Sinha

---

## [Editor Report · Acceptance letter]

28 Mar 2024

PONE-D-23-20487R1 

PLOS ONE

Dear Dr. Pulvirenti, 

I'm pleased to inform you that your manuscript has been deemed suitable for publication in PLOS ONE. Congratulations! Your manuscript is now being handed over to our production team.

Kind regards, 

on behalf of

Dr. Chen Li 

Academic Editor

PLOS ONE